# Full-Length Transcriptomic Sequencing and Temporal Transcriptome Expression Profiling Analyses Offer Insights into Terpenoid Biosynthesis in *Artemisia argyi*

**DOI:** 10.3390/molecules27185948

**Published:** 2022-09-13

**Authors:** Ran Xu, Yue Ming, Yongchang Li, Shaoting Li, Wenjun Zhu, Hongxun Wang, Jie Guo, Zhaohua Shi, Shaohua Shu, Chao Xiong, Xiang Cheng, Limei Wang, Jingmao You, Dingrong Wan

**Affiliations:** 1School of Life Science and Technology, Wuhan Polytechnic University, Wuhan 430023, China; 2College of Osteopathic Medicine, Kansas City University of Medicine and Biosciences, Joplin, MO 64804, USA; 3Institute of Chinese Herbal Medicines, Hubei Academy of Agricultural Sciences, Enshi 445000, China; 4Key Laboratory of Chinese Medicine Resources and Compound Formula, Ministry of Education, Hubei University of Chinese Medicine, Wuhan 430065, China; 5College of Plant Science and Technology, Huazhong Agricultural University, Wuhan 430070, China; 6School of Pharmaceutical Sciences, South-Central University for Nationalities, Wuhan 430074, China

**Keywords:** *Artemisia argyi*, SMRT sequencing, temporal transcriptome, gene expression patterns, terpenoid biosynthesis

## Abstract

*Artemisiae argyi* Folium is a traditional herbal medicine used for moxibustion heat therapy in China. The volatile oils in *A.*
*argyi* leaves are closely related to its medicinal value. Records suggest that the levels of these terpenoids components within the leaves vary as a function of harvest time, with June being the optimal time for *A. argyi* harvesting, owing to the high levels of active ingredients during this month. However, the molecular mechanisms governing terpenoid biosynthesis and the time-dependent changes in this activity remain unclear. In this study, GC–MS analysis revealed that volatile oil levels varied across four different harvest months (April, May, June, and July) in *A. argyi* leaves, and the primarily terpenoids components (including both monoterpenes and sesquiterpenes) reached peak levels in early June. Through single-molecule real-time (SMRT) sequencing, corrected by Illumina RNA-sequencing (RNA-Seq), 44 full-length transcripts potentially involved in terpenoid biosynthesis were identified in this study. Differentially expressed genes (DEGs) exhibiting time-dependent expression patterns were divided into 12 coexpression clusters. Integrated chemical and transcriptomic analyses revealed distinct time-specific transcriptomic patterns associated with terpenoid biosynthesis. Subsequent hierarchical clustering and correlation analyses ultimately identified six transcripts that were closely linked to the production of these two types of terpenoid within *A. argyi* leaves, revealing that the structural diversity of terpenoid is related to the generation of the diverse terpene skeletons by prenyltransferase (TPS) family of enzymes. These findings can guide further studies of the molecular mechanisms underlying the quality of *A. argyi* leaves, aiding in the selection of optimal timing for harvests of *A. argyi*.

## 1. Introduction

*Artemisia argyi* is among the most common *Artemisia* species, growing as a native plant species in China, Japan and the Korea. The leaves of *A. argyi* are often used in the traditional Chinese medicine (TCM) practice of moxibustion, for the treatment of diarrhea, tuberculosis, eczema, hemostasis, and menstruation-associated symptoms [1]. The dried leaves of *A. argyi* are often used as a food ingredient, owing to their delicious flavor and characteristic smell [2,3,4].

“Chinese Mugwort” (*A. argyi*) is the most widely used medicinal species in China. Owing to its importance as a traditional medicinal plant, ancient Chinese customs suggest that its buds and leaves be harvested before and after 5 May of the lunar calendar (usually in early June), respectively, for medicinal use. Recent analyses have similarly shown that June may be the optimal time for *A. argyi* harvesting, owing to the high levels of pharmacologically active volatile oils, tannins, and phenolics within plant tissues at this time point, which may be associated with improved antitumor, antiviral, antimicrobial, anti-inflammatory, and immunomodulatory properties [5]. In addition, dried and ground *A. argyi* leaves are the original material for moxa floss, which is used for moxibustion as a TCM therapeutic to cure dysmenorrhea, diarrhea, and fatigue, and the *A. argyi* leaf’s volatile oils play a significant therapeutic role inmoxibustion [6,7]. So, the total volatile oil content in *A. argyi* samples is often assessed as a measure of the quality of this herb, given that these oils are the major active ingredients.

The volatile oils derived from *A. argyi* leaves are primarily composed of monoterpenes and sesquiterpenes [8,9], both of which are members of a large and structurally diverse terpenoid family derived from conjugations of five-carbon dimethylallyl diphosphate (DMAPP) and its isomer isopentenyl diphosphate (IPP). Prenyl transferases are enzymes responsible for catalyzing DMAPP condensation with multiple units of IPP in a head–tail orientation, yielding prenyl diphosphate chains of varying lengths. Terpene synthases (TPSs) and cytochrome P450 (CYP) family members subsequently rearrange these compounds and drive terpenoid diversification, leading to the production of thousands of different isoprenoids. For example, monoterpene synthase and sesquiterpene synthase can generate monoterpenes and sesquiterpenes from geranyl diphosphate (GPP) and farnesyl diphosphate (FPP) precursors [10,11], respectively.

While the levels of these different terpenoids within *A. argyi* leaves vary as a function of harvest time, the molecular basis for these components’ time-dependent changes remains elusive. As such, the present study employed a gas chromatography–mass spectrometry (GC–MS) approach to assess differences in the *A. argyi* leaf’s volatile-oil-accumulation patterns during different harvest months (April, May, June, and July). This analysis was conducted in parallel with single-molecule real-time (SMRT) sequencing and Illumina RNA-sequencing (RNA-Seq) analyses of *A. argyi* leaf samples collected at these different harvest time points, in order to clarify the underlying transcriptomic profiles in these leaves. The underlying metabolic pathways associated with the terpene biosynthesis in these samples were identified through Kyoto Encyclopedia of Genes and Genomes (KEGG) pathway analyses, and the differentially expressed genes (DEGs) associated with monoterpene and sesquiterpene biosynthesis during different harvest months were further evaluated. We additionally conducted preliminary screening for key terpenoid-biosynthesis-related candidate genes, through Pearson’s correlation analyses of the total volatile oil contents and gene expression patterns in different leaf samples. Therefore, this study provides a robust foundation for future studies of terpenoid synthesis in *A. argyi* leaves, potentially guiding genetic alteration efforts aimed at improving the yield of this culturally and economically important medicinal plant species.

## 2. Results

### 2.1. Assessment of A. Argyi Leaf Volatile Oil Contents As a Function of Harvest Time

An initial GC–MS analysis of volatile compounds identified 60 compounds from an *A. argyi* leaf sample (Appendix A), corresponding to 70% of the volatile oil content and consisting of a complex mixture of monoterpenes (58.3%), sesquiterpenes (28.3%), and other components. Ten volatile oil compounds were identified in *A. argyi*. The primary monoterpenes included O-Cymene; 1,8-Cineole; (1R)-4-methyl-1-propan; γ--Terpinene; Sabinene; Thujone; and α-Terpineol, and the primary sesquiterpenes compounds were β-Caryophyllene, β-Farnesene, and β-Selinene (Figure 1A,B and Appendix A).

We further compared the levels of these volatile oil components in *A. argyi* leaves as a function of harvest time, by selecting three replicate samples at four time points from April to July in the same year. The results indicated that monoterpene and sesquiterpene levels varied over time in the analyzed *A. argyi* leaf samples. Six of the highest volatile oil levels in *A. argyi* leaves were from samples harvested in June, while two were observed in April and one each was observed in May and July. Total levels of monoterpenes and sesquiterpenes initially rose to peak levels of 23.1% and 8.2% in early June, respectively, before falling by about 17.7% and 6.2% in July (Figure 1C,D). This suggested that the volatile oil accumulation was closely associated with the developmental stage of individual *A. argyi* plants.

### 2.2. SMRT and Illumina RNA-Seq Analyses of A. Argyi Leaf Samples

Next, we performed RNA-Seq-based transcriptomic profiling to generate a de novo transcriptomic assembly. A representative full-length transcriptome was generated by pooling equimolar amounts of DNA from four leaf samples from different collection months, prior to SMRT sequencing. A Pacific RSII sequencing instrument was used to sequence differently sized full-length cDNA libraries (1–2 kb, 2–3 kb, and 3–6 kb), ultimately generating 63,281,536 subreads (84.81 Gb) in one SMRT cell with an average read length of 1267 bp and an N50 of 1571 bp (Appendix A). In total, 888,748 read of inserts (ROIs) were generated, among which 855,813 harbored two primers and a poly-A tail and were, thus, considered to be full-length ROIs. These were further subdivided, with 740,702 being identified as full-length non-chimeric (FLNC) reads at an average read length of 1268 bp (Appendix A and Appendix A).

As SMRT sequencing can have high error rates, iterative clustering for error correction (ICE) analyses were conducted, with transcripts being clustered together to predict consensus full-length non-chimeric read of insert (FLNC-ROI) isoforms. In total, 241,963 consensus isoform sequences were identified with different sizes (0–1 kb (24,306), 1–2 kb (33,989), 2–3 kb (9304), and > 3 kb (1579)), and the mean length of these sequences was 1395 bp (Appendix A). Next, cDNA libraries were prepared from the same samples utilized for SMRT sequencing, with a deep RNA-seq being performed with an Illumina Hiseq X-ten instrument. This analysis yielded 103.04 Gb of clean reads that were used to further correct SMRT transcripts. Following error correction, 69,178 unigenes with an N50 length of 1619 bp were successfully identified from the 241,963 mRNAs, of which 46,946 unigenes were longer than 2000 bp. Relative to the assembled unigenes obtained in prior NGS analyses of *A. argyi* [11], the number of long-length unigenes in our dataset was greatly increased. Then, using BLAST, the SwissProt database was compared with Hmmscan to search for Pfam domain homology as a means of predicting coding regions. A total of 69,178 complete transcripts were obtained (Appendix A). The details of the prediction results are listed in Appendix A, and the coding sequence (CDS) length distributions are shown in Appendix A.

### 2.3. Functional Annotation

To more fully understand the mechanisms governing terpenoid biosynthesis in *A. argyi*, functional annotation of FLNC transcripts was conducted using a series of databases. In total, 69,178 unique isoforms were annotated with BLASTX (v 2.2.26) and the COG (Clusters of Orthologous Groups) [12], GO [13], KEGG [14], KOG [15], Pfam (Protein family) [16], Swissprot [17], and NR [18] databases (Figure 2A, Appendix A).

GO functional annotations were used to assign specific biological process (BP), molecular function (MF), and cellular component (CC) terms to the unique isoforms, yielding three primary GO categories and 56 subcategories. A large proportion of these genes were annotated in the BP terms (“cellular process”, “metabolic process”, and “biological regulation”), the CC terms (“membrane”, “membrane part”, and “cell”), and the MF terms (“catalytic activity”, “binding”, and “transcription regulator activity”) (Figure 3C).

In total, these genes were associated with 137 KEGG pathways, many of which were related to growth and metabolite accumulation (Figure 2B).

### 2.4. Individual Transcriptomic Analyses of A. Argyi Leaf Samples Collected at Four Different Developmental Time Points

Next, cluster dendrogram and principal component analysis (PCA) analyses were conducted using samples from different *A.*
*argyi* developmental stages. Samples from these four *A. argyi* developmental stages were clustered into four separate groups, and the samples from the same collection month were assembled into the same branches, based on the results of the clustering tree, revealing clear differences in gene expression profiles among these samples (Figure 3A). PCA results revealed that the samples from May and April were significantly different from those collected in June and July, based upon the PC1 axis (Figure 3B), which is consistent with the results observed in the clustering dendrogram.

The time-specific expression of genes was further assessed to reveal differences in the transcripts from the *A. argyi* samples from four different time points. In total, 67,117 unigenes were applied to create gene intersection profiles. A total of 52,867 unigenes were shared among *A. argyi* from the four harvest time points, with 1006, 658, 536, and 616 unigenes being specifically expressed in April, May, June, and July, respectively (Figure 4).

### 2.5. Identification and Enrichment Analyses of DEGs Associated with Particular A. Argyi Developmental Stages

To further explore patterns of differential gene expression across these *A. argyi* samples collected at the four selected harvest time points, we employed fold change (FC) and FDR values to compare the expression among samples [11], leading to the identification of 140,611 DEGs, with individual comparisons among samples being made to identify DEGs that were up- or down-regulated. Specifically, DEGs were identified by comparing samples from May and April, April and June, April and July, May and June, May and July, and June and July. The greatest number of DEGs (33,968) was observed when comparing samples from April and July, while the smallest number (15,351) was evident when comparing samples from June and July (Appendix A), consistent with the important roles of many of these genes during the different stages of *A. argyi* development.

In a soft clustering analysis, 12 gene clusters were obtained with distinctive characteristics. For example, clusters 10 and 12 exhibited trends toward consistent up-regulation and down-regulation over time, respectively. The other 10 clusters exhibited varying patterns of expression changes over time from April through July. Notably, gene expression in clusters 3 and 5 tended to rise from April to June and to decrease in July; these genes, thus, exhibited trends consistent with the overall trend in volatile oil accumulation at these harvest time points (Figure 5). As such, these two clustering modules were considered to be correlated with *A. argyi* terpenoid synthesis.

### 2.6. Identification of Terpenoid-Biosynthesis-Related Genes

Through the integrated annotation approach outlined above, 77 unigenes were annotated as being associated with terpenoid backbone biosynthesis, suggesting that similarity-based BLASTX searches conducted with the Swiss-Prot database were more powerful than other tested analytical approaches. In order to assess the presence of these transcripts in other species, we conducted BLASTX analyses against a collection of plant cDNAs from protein-coding genes. We were able to identify 68 (88.3%) transcripts with at least one significant match, 17 of which exhibited high similarity to *Artemisia annua.* All unigenes with >500 bp in length, meeting functional annotation criteria, were ultimately evaluated, leading to the identification of 77 terpenoid biosynthesis-related unigenes, 44 of which were full-length cDNA sequences (Appendix A).

To clarify the potential biological functions of these genes, hierarchical clustering analyses were next performed based upon their log-2-transformed FPKM values in the analyzed *A. argyi* leaf samples. Ultimately, these genes were assigned to four distinct clusters. Group I contained 11 genes, including 1 gene in the MVA pathway (HMGR), 5 genes in the MEP pathway (1 HDS, 2 DXR, and 2 HDR), and 5 prenyltransferases (4 FPS, and 1 FPPS). These group I genes were expressed at the highest levels in *A. argyi* samples collected in May. In total, 29 genes were assigned to group II, including 9 genes in the MVA pathway (2 AACT, 3 MVK, 2 MDD, 1 PMK, and 1 HMDS), 15 genes in the MEP pathway (1 CMK, 9 DHR, 1 MVK, 2 PMK, 1 HDS, and 1 MCT), and 5 prenyltransferases (5 FPPS). These group II genes were expressed at the highest levels in samples collected in June or July. Group III contained 1 HMGS gene in the MVA pathway and 10 MEP pathway genes (2 HDS, 3 CMT, 1 CMK, 3 MDS, and 1 HDR) that were expressed at higher levels in the April or July samples. Lastly, 26 genes were assigned to group IV, including 1 HMGR gene in the MVA pathway, 15 MEP pathway genes (6 HDS, 2 MCT, 1 DXR, 5 MDS, and 1 HDR), and 10 prenyltransferases (2 IPPI, 2 FPPS, and 6 GPPS) that were expressed at the highest levels in the April samples (Figure 6 and Appendix A).

### 2.7. Assessment of Correlations between Terpenoid Content and Gene Expression

To better understand the regulation of volatile oil biosynthesis, transport, and accumulation over time in *A. argyi* leaves, we conducted further analyses of the genes associated with terpenoid backbone biosynthesis and modification. Eleven of these genes (from Figure 6) were maximally expressed in June, when these leaf samples exhibited peak volatile oil levels. Pearson correlation analyses revealed these genes to be significantly correlated with the terpenoid content in these leaf samples (Figure 7). In particular, the expression of HMGR1 from the MAV pathway was significantly correlated with 1,8-Cineole, (1R)-4-methyl-1-propan, γ--Tepinene, Sabinene, Thujone, and β-Selinene contents, while MDD1 and MDD2 were correlated with α-Terpineol levels. The HDR1 and HDR3 genes of the MEP pathway were also significantly correlated with O-Cymene, 1,8-Cineole, and β—Farnesene content, while HDS7 was significantly correlated with the γ-Terpinene, Sabinene, and β-Selinene levels. In addition, TPS2 was significantly positively correlated with the levels of (1R)-4-methyl-1-propan contents, and TPS5 was significantly positively correlated with the levels of the O-Cymene, Thujone, and β—Farnesene contents, respectively. These results identified six key genes: one 3-hydroxy-3-methylglutaryl-CoA reductase (HMGR1), one (E)-4-hydroxy-3-methylbut-2-enyl diphosphate synthase (HDS7), two 1-deoxy-D-xylulose-5-phosphate reductoisomerase (HDR1, HDR7), and two terpene synthase (TPS2,TPS5) genes that were significantly correlated with the levels of monoterpenes or sesquiterpenes in the *A. argyi* samples. The structural diversity of the terpenoids is related to the generation of diverse terpene skeletons by TPS [19], and these latter two enzymes are likely to regulate monoterpene and sesquiterpene distributions over the course of *A. argyi* leaf development. RT-qPCR results revealed that these six gene expression profiles were generally consistent with the FPKM values (Figure 8).

## 3. Discussion

### 3.1. Harvest Timing Impacts the Quality and Medicinal Value of A. Argyi Leaves

The quality of medicinal materials is often associated with environmental parameters and other cultivation-related factors. Harvest timing, production areas, and germplasm resources are all known to be key determinants of the levels of bioactive compounds within a given plant [20]. In the present study, the quality of *A. argyi* leaves was analyzed based upon the volatile oil contents therein, revealing that these volatile oil levels (primarily composed of monoterpenes and sesquiterpenes) initially rose and then fell over the course of collection. Peak volatile oil levels were observed for samples collected in early June. This is in line with other reports, suggesting that *A. argyi* leaves harvested in June are often of higher quality [21,22,23]. These results also largely agree with ancient Chinese customs. Therefore, our current study reaffirmed harvest timing as the most important factor influencing the volatile oil contents within *A. argyi* leaves. Other factors not analyzed in this study, such as irrigation or fertilization strategies, can also influence the biochemical makeup of these herbal specimens and warrant further study.

### 3.2. PacBio Iso-Seq Sequencing and Annotation of A. Argyi

*A. argyi* is a member of the *Asteraceae* family, which is composed of over 500 species, [24] and exhibits extensive genomic complexity that has largely been unexplored to date. Most prior studies of the *A. argyi* genome were conducted based on Illumina RNA-Seq data or predictive analyses, which are well-suited for capturing coding information within mRNAs but are likely to overlook sequence information within untranslated regions (UTRs). In this study, we identified 855,813 full-length non-chimeric *A. argyi* transcript sequences via a PacBio SMRT sequencing approach, revealing that the lengths of the 5′- and 3′-UTR of most *A. argyi* mRNAs were roughly 150 and 300 nucleotides, respectively, consistent with those of Arabidopsis, tomato, and soybean analyzed via Sanger sequencing [25,26,27], confirming the high quality of such PacBio -based full-length cDNA sequencing.

Relatively few full-length transcripts are derived from Illumina RNA-Seq assemblies, and the rates of inaccurate gene structure characterization can be relatively high as a consequence of misassembly. This problem is particularly pronounced for species lacking an available reference genome [28]. SMRT Sequel is a recently developed third-generation sequencing (TGS) approach using the PacBio sequencing platform. Non-assembled long-read transcripts can be generated through this approach with a low error rate (10%), and the remaining error can be largely eliminated through RNA-seq-mediated correction [29]. Herein, we observed an *A. argyi* transcript error rate of under 1%, and, in another study, the mapping rate for maize long reads rose from 11.6% to 99.1% following Illumina correction [30]. Notably, these full-length data enabled the annotation of *A. argyi* mRNA UTRs and the construction of more accurate gene models, especially for genes expressed at lower levels and genes with long intronic regions. Overall, our data offer new insights into the *A. argyi* transcriptome, providing a valuable foundation for future studies of gene regulation in *A. argyi*.

### 3.3. Assessment of Differential Gene Expression in A. Argyi

When DEGs were identified by comparing *A. argyi* samples collected at four time points, the greatest number was observed between April and July (33,986 DEGs), while the smallest number was observed between June and July (15,351 DEGs). This suggests that many of these DEGs may play important roles in *A. argyi* growth and development in a temporal manner. Additionally, these genes were clustered into 12 clusters via a soft clustering approach, with clusters 3 and 5 being closely correlated with the volatile oil levels in *A. argyi* samples. These results were consistent with our prior data, which demonstrated the temporal patterns of unigenes expression in these plants.

A total of 1756 sequences within our dataset were associated with secondary metabolite biosynthesis, with 77 being specifically annotated as being associated with terpenoid metabolism. A majority of the carbohydrate-metabolism-related sequences were identified in *Artemisia annua* [31]. Overall, these annotations and transcriptomic differences represent a valuable resource for the study of biosynthetic activity as a function of harvesting and developmental time in *A. argyi*.

### 3.4. Terpenoid Synthesis and Transport in A. Argyi Leaves

Prior studies have primarily employed genetic or biochemical approaches to explore the mechanisms governing terpenoid biosynthesis in *A. argyi*. Generally, terpenoid biosynthesis relies on DMAPP and IPP as precursors that are generated by the MVA and methylerythritol phosphate (MEP) pathways [7,32], with the MEP pathway being primarily involved in generating isoprenoids for the synthesis of monoterpenes [33], diterpenes [34], and carotenoids [35]. This suggests that the MEP pathway may enhance the accumulation of volatile oils by increasing the supply of precursors compounds in *A. argyi*.

HMGR is the first enzyme in the MAV pathway that plays an essential regulatory role [36,37]. In *Ginkgo biloba*, for example, the expression of this gene is correlated with terpene accumulation [13], and similar results have been observed in *Solanum tuberosum* and *Glycyrrhiza uralensis* [38,39]. The HDS and HDR genes function in the final two steps of the MEP pathway; HDS has previously been studied in *Solanum lycopersicum*, *Hevea brasiliensis*, *Arabidopsis thaliana*, and *G. biloba* [40,41,42,43], functioning as a regulator of the carbon flux of the MEP pathway and plant defense mechanisms [37], with HDR being of particular importance [44]. HDR regulates IPP and DMAPP biosynthesis in the context of andrographolide production. Herein, we conducted SMRT sequencing of *A. argyi* samples at different developmental stages and found that HDS7, HDR1, and HDR3 expression levels were positively correlated with the total volatile oil levels in analyzed leaves. As such, these genes may play an essential role in regulating the MEP pathway flux in *A. argyi*. The unigene expression levels of the MEP-pathway-related genes were higher than those for the MVA-pathway-related genes, which is consistent with the high levels of monoterpenes. Crosstalk between these two different terpene skeleton pathways has been documented, and the relative contributions of each pathway to the process of terpene biosynthesis remains uncertain [45].

The expression patterns of the TPS2 and TPS5 genes can regulate the distribution of monoterpene- and sesquiterpene-type terpenoids in the *A. argyi* samples collected during different periods. Many TPS are prolific enzymes that can produce mixtures of different proportions of the same compounds [46]. Sequence similarity alone cannot predict the specific biochemical function of a single TPS family member, because only a few amino acids can cause dramatic changes in the terpenoid structures produced by a given TPS enzyme [47,48]. Therefore, it is difficult to judge the specific functions of genes based on sequence similarity alone. The specific functions of these two genes, thus, require further verification. The future functional validation and utilization of these monoterpenoids should focus on these genes.

## 4. Materials and Methods

### 4.1. Plant Materials

*A. argyi* plants used in this study were planted in the botanical garden of Wuhan Polytechnic University (Wuhan, China), which is located at approximately N30.58, E114.03). Weeding was performed by hand, and micro-sprinkler systems were used for irrigation to support plant growth. No fungicides or pesticides were applied, as no relevant diseases or pests were present at the study site during the study period.

*A. argyi* leaves were collected monthly during the first week of April, May, June, and July from plants. Three replicate leaves were collected at each of these four time points (12 total samples), with each sample being separated in two. One half of each sample was frozen with liquid nitrogen and stored at −80 °C prior to transcriptomic analyses, while the other half was stored at −80 °C prior to measurements of the volatile oil contents.

### 4.2. GC–MS Analysis

#### 4.2.1. Isolation and Concentration of Volatiles

To measure volatile oil contents, leaf samples were ground to yield a powder that was then dried to a constant weight at 25 °C in an oven. Next, 1 g (1 mL) of the powder was transferred to a 20 mL headspace vial (Agilent, Palo Alto, CA, USA), containing a NaCl-saturated solution to inhibit any enzyme reaction. The vials were sealed using crimp-top caps with TFE-silicone headspace septa (Agilent, Palo Alto, CA, USA). At the time of solid-phase micro extraction (SPME) analysis, each vial was placed at 100 °C for 5 min, then a 120 µm divinylbenzene/carboxen/polydimethylsilioxan fiber (Agilent, Palo Alto, CA, USA) was exposed to the headspace of the sample for 15 min at 100 °C.

#### 4.2.2. GC–MS Conditions

After sampling, desorption of the VOCs from the fiber coating was carried out in the injection port of the GC apparatus (Model 8890; Agilent, Palo Alto, CA, USA) at 250 °C for 5 min in splitless mode. The identification and quantification of VOCs was carried out using an Agilent Model 8890 GC and a 5977B mass spectrometer (Agilent, Palo Alto, CA, USA), equipped with a 30 m × 0.25 mm × 0.25 μm DB-5MS (5% phenyl-polymethylsiloxane) capillary column. Helium was used as the carrier gas at a linear velocity of 1.2 mL/min. The injector temperature was maintained at 250 °C, with the detector at 280 °C. The oven temperature was programmed from 40 °C (3.5 min), increasing at 10 °C/min to 100 °C, at 7 °C/min to 180 °C, and at 25 °C/min to 280 °C, followed by a 5 min hold. Mass spectra were recorded in electron impact (EI) ionization mode at 70 eV. The quadrupole mass detector, ion source, and transfer line temperatures were set, respectively, at 150 °C, 230 °C, and 280 °C. Mass spectra were scanned in the m/z 50–500 amu range at 1 s intervals. Volatile compound identification was achieved by comparing the mass spectra with the data system library (Metware company-built GC–MS database (MWGC) or National Institute of Standards and Technology (NIST)) and linear retention index.

### 4.3. Transcriptomic Analysis

#### 4.3.1. RNA Preparation

Samples from each plant were ground on dry ice, after which the CTAB-PBIOZOL reagent and ethanol precipitation were employed to purify total RNA from these samples based on provided instructions. RNA quality was assessed with an Agilent 2100 bioanalyzer (Thermo Fisher Scientific, Waltham, MA, USA) and an RNA 6000 Nano Labchip kit (Agilent Technologies, Santa Clara, CA, USA), and samples exhibiting an RNA integrity number between 7 and 10 were used for RNA-seq analysis, as described by Jin et al. [49].

#### 4.3.2. PacBio Library Construction and Sequencing

Equal portions of the four total RNA samples obtained from 4 collection time points (April, May, June, and July) were pooled to produce a mixed RNA sample for PacBio sequencing. The Iso-Seq Template Preparation for Sequel Systems protocol was then employed to prepare sequencing templates. A total of 800–1000 ng of total pooled RNA was then utilized for the first-strand cDNA synthesis with the Clontech SMARTer PCR cDNA Synthesis Kit, with CDS Primer IIA first being annealed to the polyA+ tail of transcripts, after which the SMARTScribe Reverse Transcriptase was employed for first-strand synthesis. Large-scale PCR was then used to generate sufficient cDNA using Clontech PrimeSTAR GXL DNA Polymerase and 5′ PCR Primer IIA (5′-AAGCAGTGGTATCAACGCAGAGTAC-3′). The DNA Damage Repair Mix (PacBio, Menlo Park, CA, USA) and End Repair Mix (PacBio) solutions were then used to repair the prepared cDNA, which subsequently underwent adapter ligation using an appropriate ligase (PacBio, Menlo Park, CA, USA), with Exo III and Exo VII being introduced into the solution to reduce unrepaired DNA or linear DNA lacking blunt adapter sequences. The remaining cDNA samples were then measured with Qubit HS (Life Technologies, Carlsbad, CA, USA) and Agilent 2100 Bioanalyzer (Agilent, Palo Alto, CA, USA) instruments. Next, a Sequel Binding Kit 2.1 was used to bind these prepared SMRT bell sequencing libraries to appropriate polymerases using Primers V3. A PacBio MagBead Binding Kit or diffusion loading were then used to isolate polymerase-template complexes, followed by analysis with the PacBio Sequel sequencer (BGI-Shenzhen, Shenzhen, China) using Sequel Sequencing Kit 2.1 and the Sequel SMRT Cell 1M v2 Tray. The transcriptome sequencing reads were deposited in NCBI SRA with accession number SUB10986546.

#### 4.3.3. Illumina Library Preparation and Sequencing

After RNA from each leaf sample had been pooled in equimolar amounts, they were used for Illumina library preparation. First, mRNA was purified using oligo (dT) magnetic beads, after which it was then purified and fragmented into small pieces with a fragmentation buffer. Random hexamer primers were then used for first-strand cDNA synthesis, followed by second-strand cDNA synthesis. Samples then underwent index adapter ligation, poly-adenylation, and end repair. The resultant cDNA fragments were subjected to PCR amplification and purified with Ampure XP Beads dissolved in EB solution. An Agilent Technologies 2100 Bioanalyzer was used to ensure the quality of the resultant product, and double-stranded PCR products from the prior step were heated, denatured, and circularized with the splint oligo sequence to yield a final library, which was amplified with phi29 to generate DNA nanoballs (DNBs) containing over 300 copies of a given molecule. These DNBs were then loaded into a patterned nanoarray, and single-end 50 bp reads were obtained using a BGIseq500 instrument (BGI-Shenzhen, Shenzhen, China). The transcriptome sequencing reads were deposited in NCBI SRA with accession number SUB10986546.

#### 4.3.4. Full-Length Transcriptome Analysis

There were three key stages in the full-length transcriptome sequencing performed for the present study [50]: full-length sequence recognition, isoform-level clustering to ensure sequence consistency, and consistent sequence polishing. Initially, ROI sequences were filtered based on the presence of 3′- or 5′-primer sequences and poly-A tails and whether or not they contained full-length or chimeric sequences. Full-length sequences from the same isoform were then grouped using an iterative isoform-clustering algorithm, which was used to cluster the full-length sequences from the same isoform, and similar full-length sequences were clustered together. Each cluster, thus, contained a consistent sequence. Non-full-length sequences were then clustered with the Quiver algorithm, the resultant consistent sequences underwent polishing, and then high-quality sequences were screened for subsequent analysis. Only high-quality sequences were screened, as the deletion of the 5′-end of a sequence had the potential to indicate a non-full-length sequence such that only 5′ exon sequences were pooled; the longest sequence was used as the final transcript sequence.

#### 4.3.5. DEG Identification

Read counts were normalized based on calculations of the fragments per kilobase of transcript per million mapped read (FPKM) values [51], with relative expression levels further being determined based upon false discovery rate (FDR) values. Genes differentially expressed among leaves collected during four consecutive months (April, May, June, and July) of the same year were identified with the DESeq R package (1.10.1), which analyzed digital gene expression data based on a negative binomial distribution model [52]. The resultant *p*-value thresholds were corrected for multiple testing based upon FDR values [53], with an FDR < 0.01 and a fold change ≥ 2 being used to identify DEGs in this analysis.

RT-qPCR was performed on an ABI Prism 7900 Sequence Detection System (Applied Biosystems, Warrington, UK). SYBR Green Real-Time PCR Master Mix (Takara, Japan) was used for each PCR reaction in a 20 μL reaction volume, containing 1 μL of each primer (5 mM) and 4 μL of first-strand cDNA. The primers used for RT-qPCR are listed in Appendix A.

#### 4.3.6. Functional Annotation and Metabolic Pathway Analyses

Resultant mRNA sequences were compared via a BLAST approach to the NCBI non-redundant NR ftp://ftp.ncbi.nlm.nih.gov/blast/db (accessed on 21 October 2021), Swiss-Prot http://ftp.ebi.ac.uk/pub/databases/swissprot (accessed on 21 October 2021), and clusters of euKaryotic Orthologous Groups (KOG) http://www.ncbi.nlm.nih.gov/KOG (accessed on 21 October 2021) databases to derive NR, Swiss-Prot, and KOG annotations, respectievly. Gene Ontology (GO) annotations http://geneontology.org (accessed on 21 October 2021) were established based on the closest BLASTX hit from the NR database using the WEGO software (E-value ≤ 10^−5^) [54]. KEGG http://www.genome.jp/kegg (accessed on 21 October 2021) pathway analyses were conducted with the KEGG Automatic Annotation Server (KAAS) at a threshold of E ≤ 10^−5^ [55].

## 5. Conclusions

In conclusion, we employed SMRT sequencing and RNA-seq strategies to efficiently identify key genes associated with the terpenoid biosynthesis in *A. argyi*. We evaluated temporal expression patterns of these metabolically important genes at four harvest time points, leading to the identification of six key genes, including one 3-hydroxy-3-methylglutaryl-CoA reductase (HMGR), one (E)-4-hydroxy-3-methylbut-2-enyl diphosphate synthase (HDS), two 1-deoxy-D-xylulose-5-phosphate reductoisomerase (HDR), and two terpene synthase (TPS) genes that were significantly correlated with the levels of monoterpenes or sesquiterpenes in *A. argyi* samples. Together, these data provide a robust scientific foundation for future studies exploring the molecular determinants of *A. argyi* sample quality and optimized sample harvesting, which may support genetic engineering efforts focused on the secondary metabolic pathways within relevant medicinal plants.

## Figures and Tables

**Figure 1 molecules-27-05948-f001:**
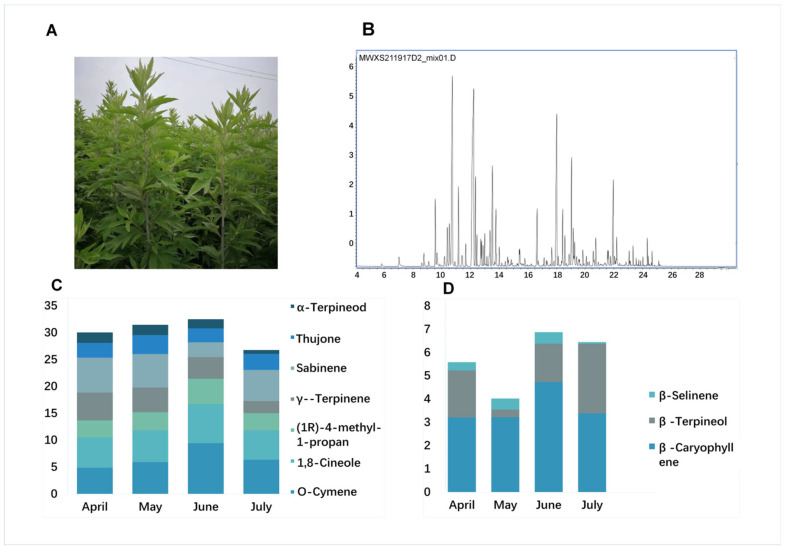
GC–MS chromatograms and terpenoid contents in *A. argyi*. (**A**) *A. argyi* plants cultivated under field conditions. (**B**) GC–MS total ion chromatograms of volatile compounds from *A. argyi* leaf samples collected in April, May, June, and July, respectively. (**C**) Bar graph corresponding to the distribution of levels of seven monoterpenes in *A. argyi* samples collected at four time points. (**D**) Bar graph corresponding to the distribution of levels of three sesquiterpenes in *A. argyi* samples collected at four time points.

**Figure 2 molecules-27-05948-f002:**
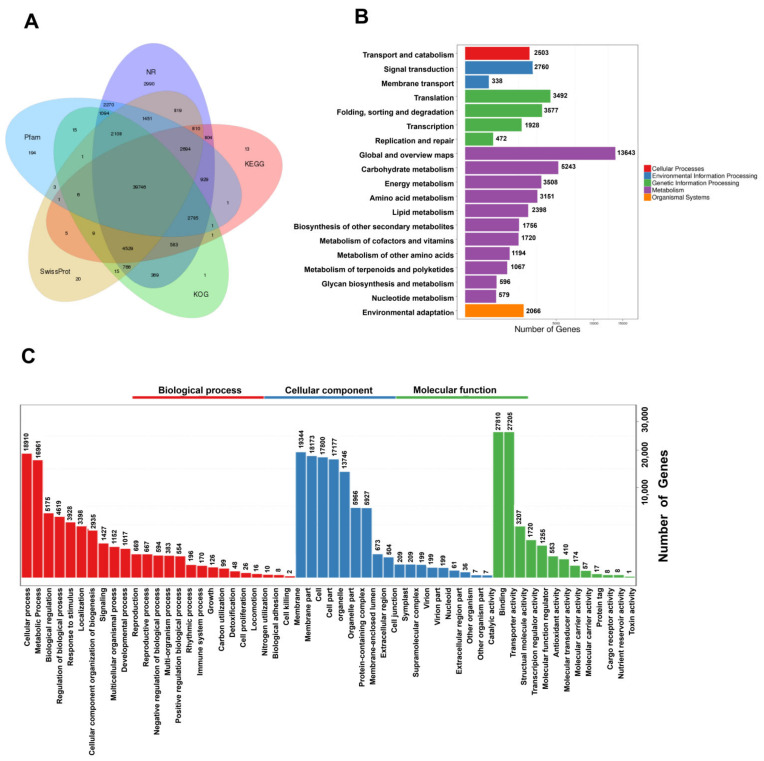
Functional annotation and classification of assembled *A. argyi* unigenes. (**A**) Annotation details arranged in a Venn diagram. (**B**) KEGG pathway classifications for putative proteins. (**C**) GO classifications for *A. argyi* unigenes, including biological process (BP), cellular component (CC), and molecular function (MF) annotations.

**Figure 3 molecules-27-05948-f003:**
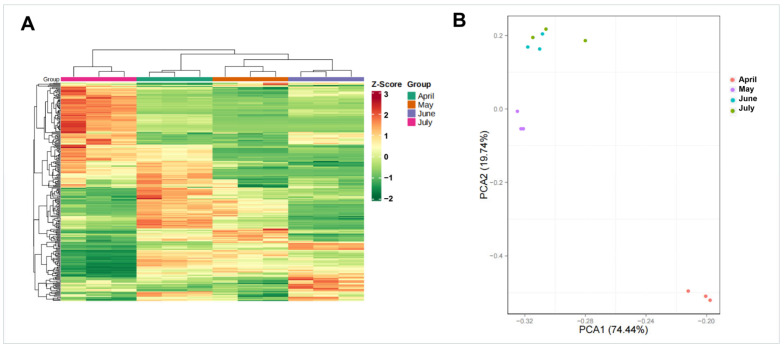
Gene expression levels during four *A. argyi* developmental stages. (**A**) Clustering dendrogram of gene expression patterns in different *A. argyi* developmental stage samples. (**B**) PCA of gene expression profiles for different *A. argyi* developmental stage samples.

**Figure 4 molecules-27-05948-f004:**
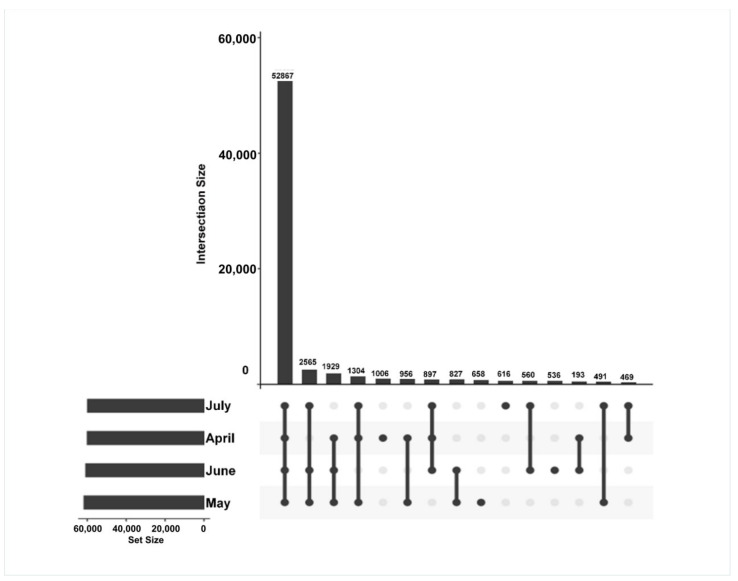
Gene intersection matrix profiles for *A. argyi* samples collected at four time points.

**Figure 5 molecules-27-05948-f005:**
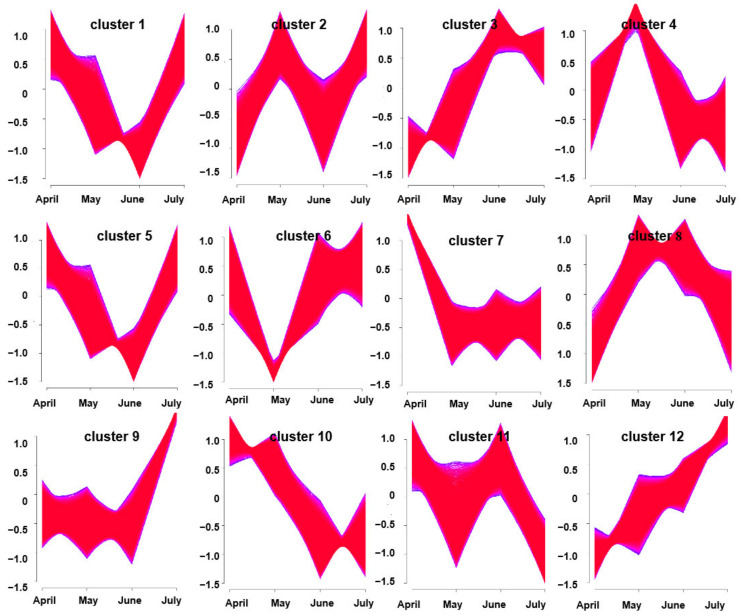
Clusters of DEGs obtained via soft clustering for *A. argyi* samples collected at four time points.

**Figure 6 molecules-27-05948-f006:**
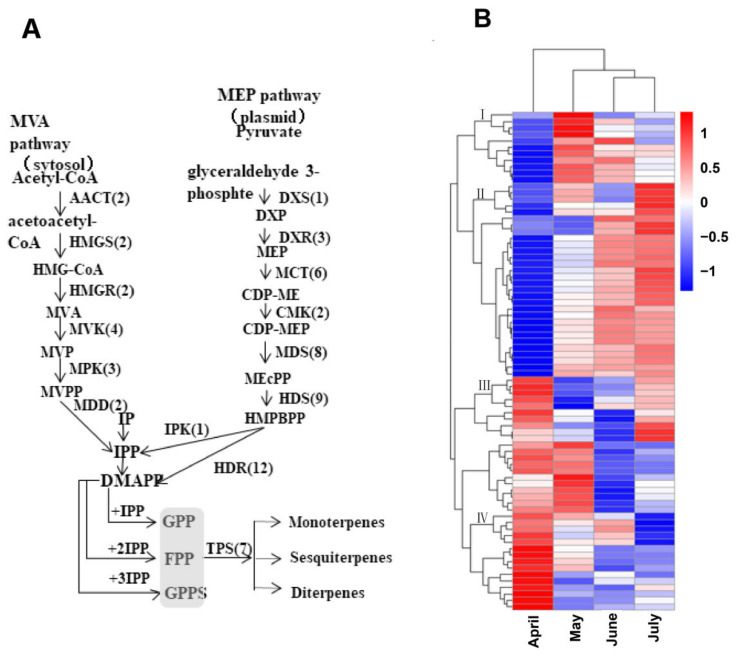
Schematic overview of the putative *A. argyi* terpenoid biosynthetic pathway, with corresponding gene expression levels. (**A**) The predicted pathway of *A. argyi* terpenoid biosynthesis, with the numbers in brackets denoting the number of enzymes in each family. (**B**) A clustered heatmap demonstrating the log-2-transformed FPKM expression levels of 77 terpenoid backbone genes and 7 TPS genes in *A. argyi* samples collected during the indicated months.

**Figure 7 molecules-27-05948-f007:**
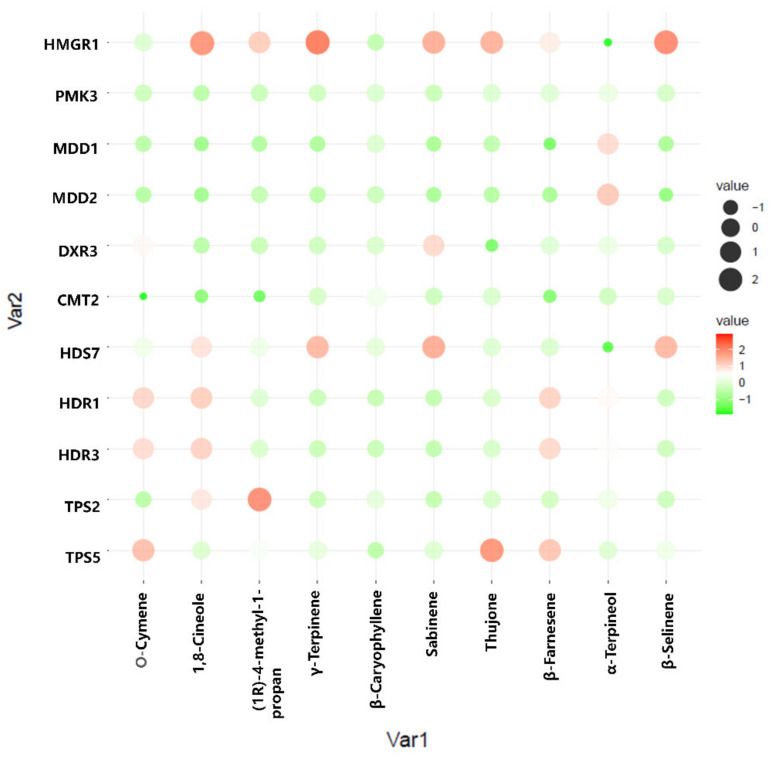
Pearson correlation bubble chart corresponding to gene expression patterns and terpenoid contents in *A. argyi* samples collected at four time points. The size of circles corresponds to correlation coefficient (R) values, and colors indicate whether a correlation is negative or positive.

**Figure 8 molecules-27-05948-f008:**
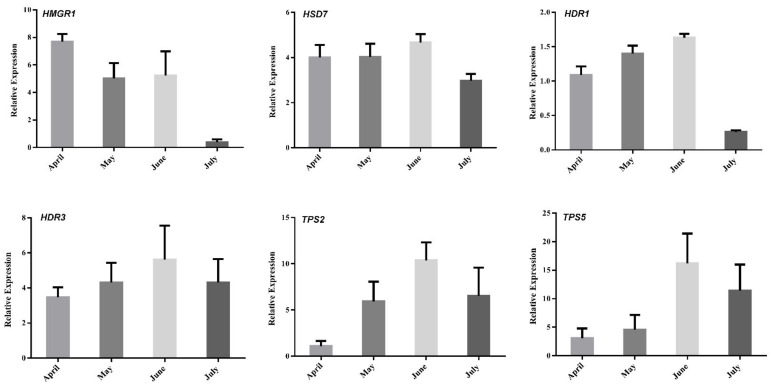
Expression analyses of selected genes (from Figure 7) using both qRT-PCR and RNA-Seq (FC of FPKM) approaches. Relative expression levels were estimated from the threshold of the PCR cycle, by the delta–delta Ct method. The error bars indicate the standard errors of three biological replicates.

## Data Availability

The data presented in this study are available on request from the corresponding author.

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
