# Peer review of "Full-Length Transcriptomic Sequencing and Temporal Transcriptome Expression Profiling Analyses Offer Insights into Terpenoid Biosynthesis in Artemisia argyi"

_molecules, 2022, doi:10.3390/molecules27185948_

Round 1

Reviewer 1 Report

The manuscript of Xu et al. reports the transcriptome analysis of Artemisia argyi leaves in different harvesting time in relation to terpenoid biosynthesis through combined approach of GC-MS and SMRT and illumina RNA-sequencing. Even though the novelty of the study is limited to only Artemisia argyi, a medicinal plant, I think it’s a good addition to the literature. The manuscript is well written for most of the part. However, there are number of issues that must be addressed prior to publication. Some of the comments or suggestions are given below-

Results:

-          The authors mentioned in 2.1 (page#2) that they identified 60 compounds by GC-MS analysis, but I don’t see those 60 compounds anywhere in the table or supplementary tables. The author may think about putting them in supplementary table. However, the authors did show 10 main compounds in figures and supplementary figures.

-          The authors mentioned that they pooled the DNA from 4 leaf samples from different collection months for SMRT sequencing. However, they did not mention any details about RNAseq samples, e.g. number of replications, whether they pool the replications or whether samples were pooled from 4 collection months (I hope they did not pool samples from 4 collection months). However, These details can go to M&M sections instead of results section.

-          I would suggest authors to put the elaboration for any abbreviation used for the first time in the text (e.g. ROI, ICE).

-          In 2.7, the description and the figure 8 do not match. For example, in L331-332, it says, “o-Cymene levels were significantly positively corelated with the levels of prenyltransferases TPS2”…whereas it is negatively corelated. Besides, some of the sentences are too long and confusing, I would suggest to rephrase those sentences (see attached manuscript).

-          The caption/titles of the figures/tables should be self-explanatory. I would suggest the authors to provide necessary details in the caption of the Figure 5, Figure 8 and Figure 9 to make the figure understandable (see attached manuscript). It would be easy to follow the text if the authors put the compounds’ full name in caption of the figure like text.

-          The authors validated only two identified key genes (TPS2 and TPS5) by qRT-PCR. I would strongly suggest to validate at least all the 8 identified key genes by qRt-PCR. Besides, the figure 9 does not tell us the relative expressions of which genes it is showing in figure A and B.

Discussions

-          Please put the references for L391-393.

-          Some sentences are too long that breaks the flow of reading e.g. L430-434. I would suggest to split the long sentence into two sentences to make it more understandable.

M&M

- I would suggest to change the subheading, “Chemical analysis” to GC-MS analysis.

- Th word ‘pulverized’ can be changed to ‘ground’.

- In 4.2.1 section, please put the references and elaborate the abbreviation ‘SPME’ (L480).

- The subsection “4.3.6 DEG identification” can be placed before 4.3.5 (see attached manuscript)

- In the conclusion the authors mentioned about identification of 8 genes but put the name of only 6 genes

- Please also look carefully at the attached manuscript to see the reviewer’s specific comments and suggestions and other corrections/typos that were not mentioned here.

I would suggest accepting the paper with major revision. I strongly encourage the authors address all the suggestions or corrections, put enough information to make the titles/captions of the figures self-explanatory and improve the flow of reading by splitting very long sentences, and resubmit a REVISED manuscript.

Author Response

Comments and Suggestions for Authors

The manuscript of Xu et al. reports the transcriptome analysis of Artemisia argyi leaves in different harvesting time in relation to terpenoid biosynthesis through combined approach of GC-MS and SMRT and illumina RNA-sequencing. Even though the novelty of the study is limited to only Artemisia argyi, a medicinal plant, I think it’s a good addition to the literature. The manuscript is well written for most of the part. However, there are number of issues that must be addressed prior to publication. Some of the comments or suggestions are given below.

Results:

1. The authors mentioned in 2.1 (page#2) that they identified 60 compounds by GC-MS analysis, but I don’t see those 60 compounds anywhere in the table or supplementary tables. The author may think about putting them in  supplementary table. However, the authors did show 10 main compounds in figures and supplementary figures.

Thank you for noticing the information that we ignored in “Results 2.1”. We have added “the identification of 60 compounds in A. argyi samples by GC-MS” in supplementary table 1.

2. The authors mentioned that they pooled the DNA from 4 leaf samples from different collection months for SMRT sequencing. However, they did not mention any details about RNAseq samples, e.g. number of replications, whether they pool the replications or whether samples were pooled from 4 collection months (I hope they did not pool samples from 4 collection months). However, these details can go to M&M sections instead of results section.

We regret not describing this experimental step in detail, and these details have been added in M&M sections 4.3.2.

3. I would suggest authors to put the elaboration for any abbreviation used for the first time in the text (e. g. ROI, ICE).

Thank you for this kind suggestion. We have supplemented the full name of the abbreviations that used for the first time in the text.

4. In 2.7, the description and the figure 8 do not match. For example, in L331-332, it says, “o-Cymene levels were significantly positively corelated with the levels of prenyltransferases TPS2”…whereas it is negatively corelated. Besides, some of the sentences are too long and confusing, I would suggest to rephrase those sentences (see attached manuscript).

We have revised this fault in 2.7 paragraph, and carefully checked the draft and revised some grammatical errors and typos.

5. The caption/titles of the figures/tables should be self-explanatory. I would suggest the authors to provide necessary details in the caption of the Figure 5, Figure 8 and Figure 9 to make the figure understandable (see attached manuscript). It would be easy to follow the text if the authors put the compounds’ full name in caption of the figure like text.

Thank you for pointing out this issue. We described compounds’ full name information in the caption of these figures like text.

6. The authors validated only two identified key genes (TPS2 and TPS5) by qRT-PCR. I would strongly suggest to validate at least all the 8 identified key genes by qRt-PCR. Besides, the figure 9 does not tell us the relative expressions of which genes it is showing in figure A and B.

Thank you for this kind suggestion. We have added six key genes by qRt-PCR experimental results, and modified the Figure 8 to show all genes name that were measured.

Discussions

1. Please put the references for L391-393.

Thank you for your suggestion. We have added the reference for “Discussion 3.1” (Xu et al., 2015; Hu et al., 2016) as Reference 22-23.

2. Some sentences are too long that breaks the flow of reading e.g. L430-434. I would suggest to split the long sentence into two sentences to make it more understandable.

We agree with the reviewer’s point, and we edited the draft accordingly.

M&M

1. I would suggest to change the subheading, “Chemical analysis” to GC-MS analysis.

We have replaced the subheading,‘Chemical analysis’ by ‘GC-MS analysis’ in M&M 4.2.

2. The word ‘pulverized’ can be changed to ‘ground’.

We have replaced the word ‘pulverized’ by ‘ground’.

3. In 4.2.1 section, please put the references and elaborate the abbreviation ‘SPME’ (L480).

We have supplemented the full name of the abbreviation ‘SPME’ in the text, in 4.2.1 section.

4. The subsection “4.3.6 DEG identification” can be placed before 4.3.5 (see attached manuscript).

Thank you for this kind suggestion. We have revised the order of M&M 4.3.6 and 4.3.5.

5. In the conclusion the authors mentioned about identification of 8 genes but put the name of only 6 genes

We apologize for the confusion caused by this paragraph. We have accordingly corrected these errors in the text.

6. Please also look carefully at the attached manuscript to see the reviewer’s specific comments and suggestions and other corrections/typos that were not mentioned here.

We have carefully checked the draft and revised many grammatical errors and typos.

Reviewer 2 Report

Dear Chief editor and Dingrong Wan:

The ms “Full-Length Transcriptomic Sequencing and Temporal Transcriptome Expression Profiling Analyses Offer Insights into Terpenoid Biosynthesis in Artemisia argyi” and Authors: Ran Xu, Yue Ming, Yongchang Li, Shaoting Li, Wenjun Zhu, Hongxun Wang, Jie Guo, Zhaohua Shi, Shaohua Shu, Chao Xiong, Xiang Cheng, Jingmao You and Dingrong Wan. This manuscript reports the transcriptome de Artemisia argyi, during four time periods, as well as the analysis of volatile compounds produced during this period, could be interesting for readers of the journal "Molecules".

The article has an appropriate design using an interesting strategy to obtain a de novo transcriptome, which is the combination of two new generation sequencing platforms, PacBio, to obtain long fragments, and Illumina, to obtain greater depth in sequencing, and in this way correct errors obtained in the first, which allows obtaining better results. Likewise, the work correlates the expression of genes with the production of metabolites to find the metabolic pathways involved in their production. Finally, the work analyzes the temporality of gene expression and production of these metabolites at 4 different times, in order to observe the changes that occur due to the stages of plant development. However, the article has several details that must be considered before its possible publication, especially I recommend the English edition language, and mainly, several errors when presenting the results, both grammatical and analytical. I would recommend major corrections before being accepted, although it could be of interest to the international scientific community.

General comments:

- I recommend language English edition.

- The manuscript has several specific errors that must be addressed.

Manuscript:

- Abstract:

Line 32: were identified.

Line 37: (HDS),

Lines 36-40: This part could be expanded in the results section.

-Introduction:

Lines 52-54: remove.

Line 65: remove therein.

Line 75: GPP and FPP?

- Results:

Line 100: Ten compounds were main identified in A. argyi. The primary monoterpene…

Lines 111-112: but 17.7 is greater than 8.2? These two statements regarding the idea are not clear.

Figure 1: C) change white color in the bar, the compound is lost.

B) could add the name to the most important peaks of the most important compounds.

Figure 2: can be deleted or sent to supplements.

Lines 183-186: A large proportion of these genes were annotated in BP: “cellular process”, “metabolic process”, and “biological regulation”; CC: the “membrane”, “membrane part”, and “cell”; and MF “catalytic activity” “binding”, and “transciption regulator activity” (Figure 3C).

Figure 3: It has misspellings in the figures that need to be corrected, for example, figure B: eevironmental to environmental.

Figure 4: Correct:

April

May

June

July

Remove legend and 3 samples.

Figure 5: samples.

It is never mentioned which genes were amplified by qPCR, neither in the methodology nor in the results, in fact in figure 9 they say the months, but not the genes that were measured.

- Materials and Methods:

Line 473: remove therein.

Line 488: 30 m?

Lines 578-582: genes that were amplified.

- Conclusion:

Line 587-588: write the name of the genes without abbreviating.

Author Response

Manuscript:

- Abstract:

1. Line 32: were identified.

Thank you for your kind suggestion. We have replaced “identified” with “were identified”.

2. Line 37: (HDS),

We have revised it.

3. Lines 36-40: This part could be expanded in the results section.

Thank you for your suggestion. We have rewritten the abstract and deleted some unnecessary description.

-Introduction:

1. Lines 52-54: remove.

We have removed it.

2. Line 65: remove therein.

We have deleted "therein" in introduction.

3. Line 75: GPP and FPP?

Thank you for this kind suggestion. We have supplemented the full name of these abbreviations that used for the first time in the text.

- Results:

1. Line 100: Ten compounds were main identified in A. argyi. The primary monoterpene…

We have revised it in results 2.1.

2.Lines 111-112: but 17.7 is greater than 8.2? These two statements regarding the idea are not clear.

We apologize for the confusion caused by this paragraph. We have accordingly corrected these errors in results 2.1.

3. Figure 1: C) change white color in the bar, the compound is lost.

  1. B) could add the name to the most important peaks of the most important compounds.

We showed the most 10 important peaks of the most important compounds and their full names in supplementary figures 1.

4. Figure 2: can be deleted or sent to supplements.

Thank you for the suggestion. We have moved figure 2 to the supplementary materials.

5. Lines 183-186: A large proportion of these genes were annotated in BP: “cellular process”, “metabolic process”, and “biological regulation”; CC: the “membrane”, “membrane part”, and “cell”; and MF “catalytic activity” “binding”, and “transciption regulator activity” (Figure 3C).

We have revised it in results 2.3.

6. Figure 3: It has misspellings in the figures that need to be corrected, for example, figure B: eevironmental to environmental.

We have accordingly corrected these errors in the figures 3B.

7. Figure 4: Correct:

April

May

June

July

Remove legend and 3 samples.

We have deleted some redundant information in Figure 4.

8. Figure 5: samples.

We have revised it in figure 5.

9. It is never mentioned which genes were amplified by qPCR, neither in the methodology nor in the results, in fact in figure 9 they say the months, but not the genes that were measured.

Thank you for pointing this out. We have modified the Figure 9 to show all genes name that were measured.

- Materials and Methods:

1. Line 473: remove therein.

We have deleted it in “materials and methods” part.

2. Line 488: 30 m?

Thank you for pointing this out. The volatile oils components of A. argyi leaves were isolated by the GC-MS method, which using the capillary column, the length of the capillary column was indeed 30m.

- Conclusion:

1. Line 587-588: write the name of the genes without abbreviating.

We have added the full name of the genes abbreviation in conclusion.

Reviewer 3 Report

This research studied molecular basis of time-dependent changes in terpenoid content in A.Argyi. If we are discussing the variation of molecular characteristics of a specific plant used in Chinese traditional medicine then this research provides an insight into the time fluctuations of volatile oil content in this plant. It has been postulated that this plant contains volatile oils which may help alleviate certain symptoms and that the content of these oils varies in different harvest times. On the other hand, if we are discussing the field of western medicine and its practice then this research does not bring any additional information. It is an interesting, well written manuscript with clear methodology and clearly presented results. This research is in line with other reports suggesting that A. argyi leaves harvested in June are often of higher quality. The addition to the existing literature is representation of the changes in DEGs expression in time. Transcriptomic patterns after creating full-length transcriptome have been revealed and the results indicated time-specific patterns associated with terpenoid synthesis. The authors should provide a detailed description of clinical use of this plant in traditional medicine in the introduction section. Were there any clinical studies performed regarding the effectiveness of this plant?

Author Response

This research studied molecular basis of time-dependent changes in terpenoid content in A.Argyi. If we are discussing the variation of molecular characteristics of a specific plant used in Chinese traditional medicine then this research provides an insight into the time fluctuations of volatile oil content in this plant. It has been postulated that this plant contains volatile oils which may help alleviate certain symptoms and that the content of these oils varies in different harvest times. On the other hand, if we are discussing the field of western medicine and its practice then this research does not bring any additional information. It is an interesting, well written manuscript with clear methodology and clearly presented results. This research is in line with other reports suggesting that A. argyi leaves harvested in June are often of higher quality. The addition to the existing literature is representation of the changes in DEGs expression in time. Transcriptomic patterns after creating full-length transcriptome have been revealed and the results indicated time-specific patterns associated with terpenoid synthesis. The authors should provide a detailed description of clinical use of this plant in traditional medicine in the introduction section. Were there any clinical studies performed regarding the effectiveness of this plant?

Thank you for pointing out this deficiency. We provided a detailed description of clinical use of this plant in traditional medicine in the introduction section. Line  49-51, and Line 61-64.